# Multi-attribute temporal descriptive methods in sensory analysis applied in food science: Protocol for a scoping review

**Michel Visalli** [1]*, **Mara Virginia Galmarini** [2,3]

**1** Centre des Sciences du Goût et de l'Alimentation, AgroSup Dijon, CNRS, INRAE, Université de Bourgogne Franche-Comté, Dijon, France, **2** Member of CONICET, Consejo Nacional de Investigaciones Científicas y Tecnológicas, Buenos Aires, Argentina, **3** Facultad de Ingeniería y Ciencias Agrarias, Pontificia Universidad Católica Argentina (UCA), Buenos Aires, Argentina

* michel.visalli@inrae.fr

## Abstract

### Background

Sensory perception is a temporal phenomenon highly present in food evaluation. Over the last decades, several sensory analysis methods have been developed to determine how our processing of the stimuli changes during tasting. These methods differ in several parameters: how attributes are characterized (intensity, dominance or applicability), the number of attributes evaluated, the moment of sample characterization (simultaneously with the tasting in continuous or discrete time, retrospectively), the required panel (trained subjects or consumers), etc. At the moment, there is no systematic review encompassing the full scope of this topic. This article presents the protocol for conducting a scoping review on multi-attribute temporal descriptive methods in sensory analysis in food science.

### Methods

The protocol was developed according to the Preferred Reporting Items for Systematic reviews and Meta-Analyses (PRISMA) extension for Scoping Reviews checklist. The research question was "how have multi-attribute temporal descriptive methods been implemented, used and compared in sensory analysis?". The eligibility criteria were defined using the PICOS (Population, Intervention, Comparator, Outcome, Study design) framework. This protocol details how the articles of the final review will be retrieved, selected and analyzed. The search will be based on the querying of two academic research databases (Scopus and Web of Science). The main topics reported in research involving sensory analyses methods will be identified and summarized in a data extraction form. This form (detailed in the protocol) will be used to report pertinent information regarding the objectives of the review. It could also be reused as a guideline for carrying out and reporting results of future research in a more standardized way. A quality appraisal process was derived from literature. It will be applied on the included articles of the review, and could also be re-used to ensure that future publications meet higher quality levels. Finally, for the sake of transparency, the limitations of the protocol are discussed.

---

**Data Availability Statement:** No datasets were generated or analysed during the current study. All relevant data from this study will be made available upon study completion.

**Funding:** The author(s) received no specific funding for this work.

**Competing interests:** The authors have declared that no competing interests exist.

# 1. Introduction

## 1.1. Background

As a result of mastication, bolus formation, contact with saliva and body temperature, food and beverage perception changes during oral processing and for this reason it is considered a temporal phenomenon. Sensory analysis aims at understanding the sensory perception of products by measuring subject responses. Thus, several temporal methods have been developed in the past 50 years trying to capture, study, describe and quantify these changes in perception. These methods and their main characteristics are listed in Table 1. One main reference is cited for each, in case the reader needs further detail and information.

The time lapse studied goes mostly from the moment in which the product gets in contact with the mouth until some seconds after swallowing. That is to say, the time it takes for a person to evaluate one intake of the product (e.g., one sip, one bite). However, food and beverage consumption has another temporality: bite after bite (or sip after sip) perception can also change due to cumulative sensory phenomenon. This type of temporality (multi-intake or full portion evaluation) was only recently studied and, even though many of the methods could be applied to this, only a few have been used.

Time Intensity (TI) [1] was the first temporal method developed. It measures (by means of a scale) the intensity of one given attribute over a continuous period of time. It was conceived as a sort of temporal version of the Quantitative Descriptive Analysis [12] but allowing only to

**Table 1. Main sensory descriptive temporal methods published in peer-reviewed articles, in chronological order.**

| Name of the method | Abbreviation | Number of attributes | Attribute generation | Variable measured | Temporal resolution | Main reference |
|---|---|---|---|---|---|---|
| **Time-Intensity** | TI | 1 | Determined before the test (by the panel leader, etc.) | Intensity | Continuous | Lee & Pangborn (1986) [1] |
| **Intensity Variation Descriptive Methodology** | IVDM | >1 | Determined before the test (by the panel leader, by consensus, etc.) | Intensity | Discontinuous | Gordin (1987) [2] |
| **Discontinuous Time-Intensity** | DTI | >1 | Determined before the test (by the panel leader, by consensus, etc.) | Intensity | Discontinuous | Clark & Lawless (1994) [3] |
| **Progressive Profile** | PP | >1 | Determined before the test (by the panel leader, by consensus, etc.) | Intensity | Discontinuous | Jack et al. (1994) [4] |
| **Dual-Attribute Time-Intensity** | DATI | 2 | Determined before the test (by the panel leader, etc.) | Intensity | Continuous | Duizer et al. (1997) [5] |
| **Temporal Dominance of Sensations** | TDS-I TDS | >1 | Determined before the test (by the panel leader, by consensus, etc.) | Dominance and intensity (TDS-I) or dominance only (TDS) | Continuous | Pineau et al. (2009) [6] |
| **Sequential Profile** | SP | >1 | Determined before the test (by the panel leader, by consensus, etc.) | Intensity | Discontinuous | Methven et al. (2010) [7] |
| **Multi-attribute Time-Intensity** | MATI | >1 | Determined before the test (by the panel leader, by consensus, etc.) | Intensity | Continuous | Kuesten et al. (2013) [8] |
| **Temporal Check All That Applies** | TCATA | >1 | Determined before the test (by the panel leader, by consensus, etc.) | Applicability | Continuous | Castura et al. (2016) [9] |
| **Attack Evolution Finish** | AEF | >1 | Determined during the test (Free-Comment) | Dominance | 3 periods (retrospective) | Visalli et al. (2020) [10] |
| **Free-Comment Attack Evolution Finish** | FC-AEF | >1 | Determined during the test (Free-Comment) | Applicability | 3 periods (retrospective) | Mahieu et al. (2020) [11] |

measure one attribute at a time. TI has long been the temporal method of reference, but it presents several limitations: measuring only one descriptor at a time, which results in halo-dumping effect [3]. Moreover, the "signature" effect [13] (evaluators have a characteristic shape of the curve) requires a higher training to reduce variability and obtain curves that respond to product characteristics and not to individual differences, resulting also in panellist fatigue [14]. All other temporal methods have been developed trying to compensate for these limitations.

As in every quantitative method, the use of scales calls for trained assessors. But, unlike other Descriptive Analysis techniques, TI requires a higher concentration since the evaluator needs to be focused on the perception and changes in intensity for the given attribute over a period of time. This higher concentration and the continuous temporal manner of the measurement requires some extra training in comparison. In addition to the extra training sessions to manage this, measuring only one attribute increases the number of sessions needed if a multi-attribute temporal description of the product is required.

Aiming at reducing the number of sessions to attempt a description with more than one attribute, Dual Time Intensity (DATI, [5]) and Multi Attribute Time Intensity (MATI, [8]) were developed. But they were not widely implemented, probably due to the difficulty of the task: quantifying different attributes over continuous scales at the same time. Discontinuous time alternatives have been proposed to simplify the process and enable the recording of intensities within a single bite or sip ("single-intake") at uniform intervals steps or at specific moments using Intensity Variation Descriptive Methodology (IVDM, [2]), Discontinuous Time-Intensity (DTI, [3]), or Progressive Profile (PP, [4]), or over repeated or consecutive consumptions ("multiple-intakes") using Sequential Profile (SP, [7]).

Methods developed after this tried to simplify the task by recording only qualitative data. Temporal Dominance of Sensations (TDS, [6]) introduced the concept of *dominance* (different from intensity) asking the subject to choose (from a given list) the sequence of dominant sensations. As a matter of fact, when TDS was first presented, panelists were also asked to rate the intensity of the chosen attributes. But, as it was soon deemed too difficult and the intensity scoring was disued. Temporal Check-All-That-Apply (TCATA, [9]) was developed as an alternative to TDS registering the presence/absence ("applicability") of all attributes along time. TCATA added the time dimension to the static Check-All-That-Apply (CATA) method [15]. After having been used with trained panels, TDS and TCATA were gradually more and more used with consumer panels allowing also to better understand preferences in addition to product description.

The newest methods, changed from the simultaneous tasting-evaluating paradigm and proposed a retrospective measure, describing the product right after tasting but taking into account the perceived temporality. Attack-Evolution-Finish (AEF, [10]) methods summarize perception as a sequence of 3 attributes corresponding to 3 subjective periods: "Attack", "Evolution" and "Finish". FC-AEF [11] mixed static Free-Comment method [16], AEF and applicability, allowing the subjects to characterize their temporal perception using their own words instead of predefined list of descriptors.

## 1.2. Rationale for conducting the review

As can be seen in Table 1, many temporal methods have been developed over the years. Despite their differences, they all aim at measuring the same phenomenon. In food science, [17] highlighted that the most important challenge for new methodologies for sensory characterization is the identification of their limitations. Although it was referring to non-temporal DA, it also applies to temporal descriptive methods. It has not been clearly established yet in which situations methodologies provide equivalent information and when their application is

**Table 2. Existing reviews related to all temporal methods in sensory science (*total number of references in each review, as reported in Scopus).**

| Reference | Type of review | Scope | Number of references* |
|---|---|---|---|
| **Cliff & Heymann (1993) [19]** | Narrative | TI | 66 |
| **Dijksterhuis & Piggott (2000) [13]** | Narrative | TI, PP | 56 |
| **Foster et al. (2011) [20]** | Narrative | TI, TDS in studies related to food oral processing | 136 |
| **Di Monaco et al. (2014) [21]** | Narrative | TDS | 43 |
| **Devezeaux de Lavergne et al. (2017) [22]** | Narrative | TI, TDS, TCATA in studies related food oral processing | 112 |
| **Schlich (2017) [23]** | Narrative | TDS and variants | 49 |
| **Fiszman & Tarrega (2018) [24]** | Narrative | TDS, in studies related to texture | 48 |

or is not recommended. [18] recently pointed out: "*Many (sensory and statistical) methods are developed and deployed, but they are rarely compared exhaustively and objectively with alternative existing methods. Why would I adopt any new method when I have something that currently (seemingly) addresses the same task in a similar way? What benefit does it bring, and is it important enough for me to bother? How can I make findings actionable to inform product design?*".

However, to date, no such exhaustive comparison of the multi-attribute temporal sensory methods exists. Indeed, the searching methodology described in this article allowed finding several reviews on the subject, but none of them addresses all the existing multi-attribute temporal descriptive methods. Moreover, as it can be observed in Table 2, there is no systematic review on the topic.

[13, 19] are obviously no longer up to date. [20, 22] mainly focus on applications in food oral processing research. [21, 23, 24] only reviewed studies related to TDS. By way of comparison, more than 350 articles (without TI) will be considered for inclusion in the review using the protocol described in this article.

Thus, the relevance of conducting a systematic review of the academic research on temporal sensory methods seems established to set guidelines based on scientific evidence. To overcome the limitations of the previous reviews, it is necessary to include all the temporal methods, with the exception of TI which is singular in the sense that it is the only one which characterizes a single attribute. As the heterogeneous nature of the studies was not amenable to a more precise systematic review, a scoping review will be carried out.

### 1.3. Objectives

The objectives of this review, in accordance with [25] are:

1. to map the scientific literature to make an exhaustive and objective inventory of the methods available for multi-attribute temporal descriptive sensory analysis of food products,

2. to clarify working definitions and inform practices in the field,

3. to summarize findings and recommendations based on (i) and (ii) and to identify research gaps in the existing literature,

4. to disseminate research findings.

   A specific focus will be accorded to methodology and articles comparing methods.

## 2. Materials and methods

The protocol was drafted in order to ensure that the scoping review will be conformed to the Preferred Reporting Items for Systematic reviews and Meta-Analyses extension for Scoping Reviews (PRISMA-ScR) Checklist [26].

This section was organized based on the five stages of a scoping review [27]:

1. identification of the research question,

2. identification of relevant databases and literature,

3. selection of articles,

4. data extraction,

5. summarization, interpretation and dissemination of the results.

Each stage is more detailed below in line with the objectives of the current scoping review.

## 2.1. Identification of the research question

**2.1.1. Main research question and sub-questions.** The main research question is: "how have multi-attribute temporal descriptive methods been implemented, used and compared in sensory analysis?"

The main research sub-questions concern:
Method and protocol

- What temporal sensory methods (and their variants) were used?

- What characterizes the method(s)?

- Which product categories were evaluated using the methods?

- What were the reported limitations, advantages and disadvantages of each method?

- How were the methods compared in terms of advantages and disadvantages?

- Are there any remaining unanswered questions related to the method and protocol implemented?

  Data analysis

- How was the data collected with these methods analyzed?

- How was performance (accuracy, discrimination, repeatability, reproducibility) measured?

- How were the methods compared in terms of their performances?

- Are there any remaining unanswered questions related to data analysis?

  Contribution to existing scientific knowledge

- In which scientific fields (food science, psychology, etc.) were these methods applied?

- What type of temporal information was obtained with these methods?

- Did this information provide additional knowledge compared to other sensory (static measures, liking, etc.) or instrumental measures?

- Was the contribution used by other researchers? (quality, number of citations)

  Demographics of the research

- What are the geographical and historical coverages of the methods?

- What are the main journals and authors implied?

- Is the scientific area FAIR? (standardization of meta-data, open access, open data, ethical, etc.)

**2.1.2. Inclusion and exclusion criteria.** The PICO(S) (Population, Intervention, Comparator, Outcome, Study design) eligibility criteria [28] for inclusion will be as follows:

**Population.** Any human panel (trained or semi-trained panelists, consumers) will be eligible for inclusion, without any limitation on its composition.

**Intervention.** Any study aiming to evaluate the sensory properties of food or drink using a multi-attribute temporal descriptive method will be relevant for inclusion. Any methodological article related to temporal data collection or statistical analysis of multi-attribute temporal descriptive methods will also be eligible for inclusion.

TI studies will be included at the identification and screening phases to present the magnitude of the use and study of the method in comparison to the multi-attribute temporal ones. However, they will not be considered thereafter for inclusion.

Studies exclusively related to temporality of preferences or emotions without consideration to product sensory descriptions will not be considered for inclusion.

**Comparators.** Studies with or without comparator will be eligible for inclusion.

**Outcomes.** Not applicable.

**Study design.** All types of study design will be eligible for inclusion.

To ensure that the articles will be available for the future readers of the review, only peer-reviewed articles having a DOI will be eligible for inclusion. In order to avoid article duplication reviews, opinion papers, congress proceedings, doctoral thesis and book chapters will be excluded. Only articles written in English will be considered.

Literature search will include published works until January 16, 2022.

## 2.2. Identification of relevant databases and literature

**2.2.1. Information sources.** The following databases were searched: Scopus, Web of Science Core Collection (WOS), and Google scholar [29]. Only Scopus and WOS were retained (see 2.2.2 for details on this choice). Furthermore, we will search the reference lists and citing articles of included studies and related systematic reviews.

**2.2.2. Search strategy.** The Peer Review of Electronic Search Strategies (PRESS) [30] checklist was used to help constructing and validating the search strategy.

First, keywords were listed according to the PICOS criteria. No keyword related to population, comparators, outcomes and study design were added. Keywords related to intervention were identified, including the names of the published temporal methods (Table 1) established based on the knowledge of the authors: "progressive profile/profiling", "sequential profile/profiling", "temporal dominance of sensations", "TDS", "temporal check all that apply", "TCATA", "T-CATA", "attack evolution finish". Although TI was outside the scope of this research, the keywords "time intensity" and "time-intensity" were added to retrieve references related to DATI and MATI, but also to potentially detect articles citing TI and to get an idea of the bibliographic volume related to this method. Keywords referring to multiple intakes were also added: "multi", "multiple", "bite", "sip", "intake". The wildcard character (*) was used (when applicable) to ensure that variations of each keyword were found. The proximity operator was also preferred (when possible) to the "AND" operator to limit the number of results of the query.

Second, Scopus and WOS were queried on title, abstract and keywords, then results were quickly screened. The test query allowed to notice that the name of the method was not always present in the title, abstract or keywords. Thus, new generic keywords related to temporal perception of sensations ("temporal", "dynamic", "perception", "description" and "sensory

analysis") were added and their combinations added to the search. The test query also allowed to identify major "subject areas" in Scopus (Agricultural and Biological Sciences; Chemistry; Psychology) and "research areas" in WOS (Food science; Chemical analytics; Psychology). They were later used to limit the number of results of the query.

Third, previous reviews related to the research question were identified (see Table 2). It allowed to check if the queries enabled to retrieve all cited references. To this end, no restriction on the type of article was added in the search. Several iterations were needed to refine the queries in WOS and Scopus. It was concluded that, with this search criteria, Google Scholar did not bring new references, and it was therefore not added as a database.

**2.2.3. Research equations.** The following research equations were retained for each database:

```
Scopus
TITLE-ABS-KEY(
        (("progressive" PRE/1 "profil*")
        OR ("sequential" PRE/1 "profil*")
        OR ("dynamic" PRE/1 "profil*")
        OR ("time" near/1 "intensity")
        OR ("time-intensity")
        OR ("temporal" AND "dominance" AND "sensation*")
        OR ("TDS")
        OR ("temporal" AND "check" AND "all" AND "that" AND "apply")
        OR ("TCATA")
        OR ("T-CATA")
        OR ("attack evolution finish")
        OR ("intensity variation descriptive methodology")
        OR ("temporal" PRE/1 "profil*")
        OR ("temporal" PRE/1 "perception")
        OR ("temporal" PRE/1 "description")
        OR ("temporal" AND "sensory analysis")
        OR ("temporal method*")
        OR ("dynamic" PRE/1 "perception")
        OR ("dynamic" PRE/1 "description")
        OR ("multi*" PRE/1 "sip*")
        OR ("multi*" PRE/1 "bite*")
        OR ("multi*" PRE/1 "intake*")
) AND (
        LIMIT-TO (SUBJAREA, "CHEM")
        OR LIMIT-TO (SUBJAREA, "AGRI")
        OR LIMIT-TO (SUBJAREA, "PSYC")
)
WOS
TS = (
        ("progressive" NEAR/1 "profil*")
        OR ("sequential" NEAR/1 "profil*")
        OR ("dynamic" NEAR/1 "profil*")
        OR ("time" NEAR/1 "intensity")
        OR ("time-intensity")
        OR ("TI")
        OR ("temporal" AND "dominance" AND "sensation*")
        OR ("TDS")
        OR ("temporal" AND "check" AND "all" AND "that" AND "apply")
        OR ("TCATA")
        OR ("T-CATA")
        OR ("attack evolution finish")
        OR ("intensity variation descriptive methodology")
```

```
                    OR ("temporal" NEAR/1 "profil*")
                    OR ("temporal" NEAR/1 "perception")
                    OR ("temporal" NEAR/1 "description")
                    OR ("temporal" AND "sensory analysis")
                    OR ("temporal method*")
                    OR ("dynamic" NEAR/1 "perception")
                    OR ("dynamic" NEAR/1 "description")
                    OR ("multi*" NEAR/1 "sip*")
                    OR ("multi*" NEAR/1 "bite*")
                    OR ("multi*" NEAR/1 "intake*")
 )
```

## 2.3. Selection of articles

Following the search, all identified records will be collated into Mendeley citation manager and duplicates removed. The process of study selection will be presented using the Preferred Reporting Items for Systematic Reviews and Meta-Analyses extension for Scoping Reviews (PRISMA-ScR) [31] as in Fig 1.

The two authors will independently screen (i) the titles and (ii) abstracts retrieved from the database search for potentially eligible studies. The full texts of these studies will be obtained and further screened for eligibility based on the inclusion and exclusion criteria. Corresponding authors will eventually be contacted to retrieve articles not available. Potential disagreements regarding eligibility will be resolved through discussion and consensus. Reasons for exclusion of full-text assessed articles will be reported in the PRISMA diagram.

The articles cited in the reviews and included articles will be manually screened to look for potential relevant missing articles. The articles citing the included articles will be retrieved using Scopus and screened in the same way. The additional references not retrieved by the research equations will be reported in the PRISMA diagram.

## 2.4. Extraction of relevant information and critical appraisal

**2.4.1. Extraction of article metadata.** The meta-data will be reported as exported from the databases.

**2.4.2. Identification of generic standards for reporting researching involving temporal sensory methods.** A strategy was defined to identify relevant data in the most exhaustive way. As no comparable review exists, it was not possible to rely on previous similar research. Many standards have been developed to report quantitative or qualitative research using specific designs, but no guideline corresponded to the designs used for the studies that will be included in the review. Indeed, most of them used quasi-experimental research designs. "Quasi-experimental research is similar to experimental research in that there is manipulation of an independent variable. It differs from experimental research because either there is no control group, no random selection, no random assignment, and/or no active manipulation." [32]. For product-oriented questions, the independent variable (the product) was most often studied using within-subject (counterbalanced or randomized) experimental designs, with no control group. For subject or method-oriented questions, factorial designs still with no control group were mostly used.

Thus, generic standards adapted for this review were identified from "Journal Article Reporting Standards for Quantitative Research in Psychology: The APA Publications and Communications Board Task Force Report" (JARS) [33]. Applicable topics were identified from JARS (from part "Information Recommended for Inclusion in Manuscripts That Report New Data Collections Regardless of Research Design"). Some topics were renamed or grouped to be more consistent with the nomenclatures used in the articles in our area of interest. The

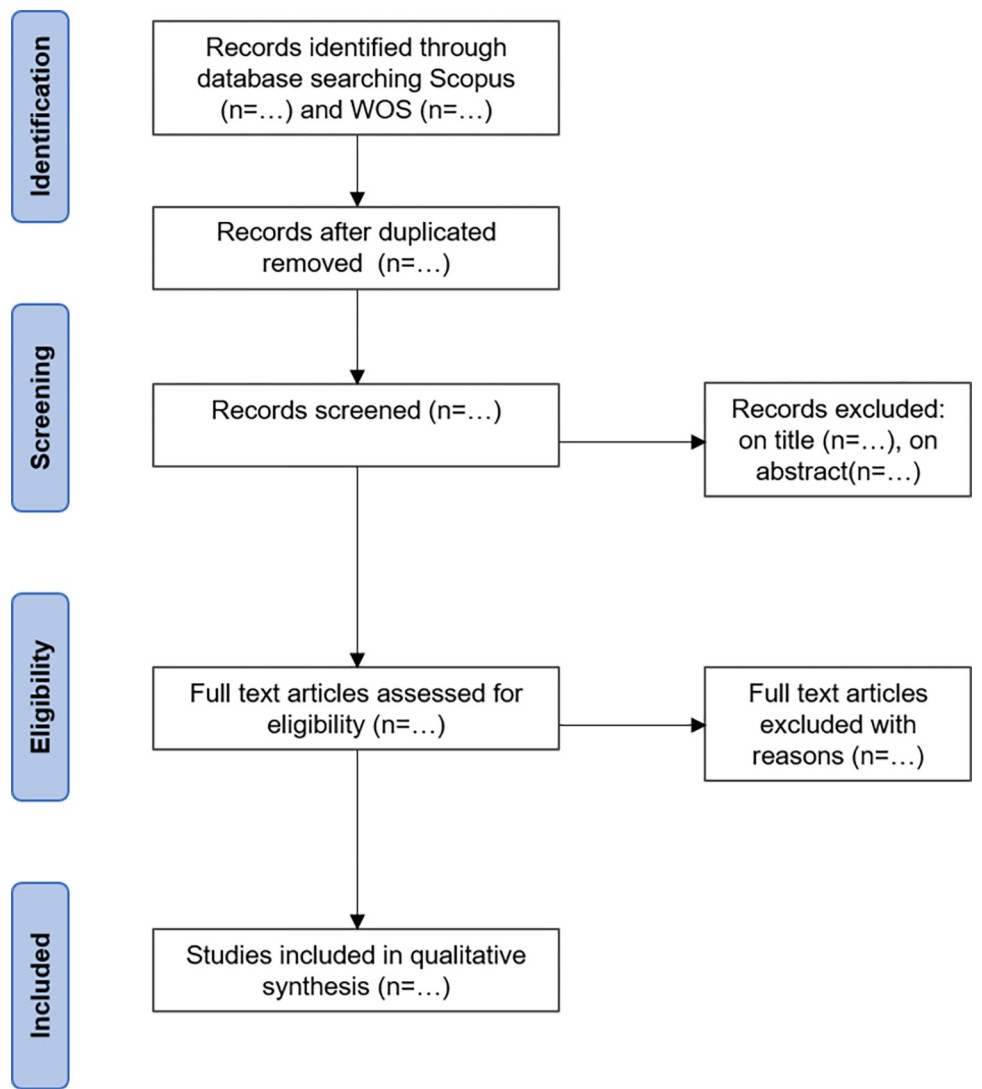

**Fig 1. PRISMA flow chart that will be completed in the final scoping review.**

expected content of each topic was completed based on literature when necessary. The result is summarized in Table 4.

**2.4.3. Definition of specific standards for reporting researching involving temporal sensory methods.** 50 articles (called hereafter "test articles") identified thanks to the database query (see 2.2.3) were randomly selected, then read. Following this reading, new topics specific to sensory analyses studies were identified. To facilitate the work of extraction following the reading of each article, a data extraction form was derived from Tables 3 and 4. To objective the topics of Table 4, the extraction form was constructed as a list of topic-related questions. When possible, the use of closed questions was preferred to maximize the agreement between the reviewers and facilitate quantitative analysis of the data in the final review. When possible, a list of pre-determined answers was suggested (the list could be extended during the final evaluation process). When the content of the information was not related to one of the identified research questions but the presence/absence of the information still relevant to be reported for

**Table 3. Meta-data exported from WOS and Scopus.**

| Meta-data | Description |
|---|---|
| DOI | Digital object identifier of the article |
| Title | Title of the article |
| Authors | Authors of the article |
| Abstract | Abstract of the article |
| Year | Year of publication of the article |
| Source title | Title of the peer-reviewed journal in which the article is published |
| Keywords | Keywords reported by the authors |
| Language | Language of the article |
| Subject/research area | Subject area of the journal |
| Number of references | Number of references cited by the article |
| Number of citations | Number of references citing this article |
| Open access to the manuscript | Type of open access |

**Table 4. Generic standards applicable for reporting studies involving temporal sensory methods, adapted from JARS.**

| Topic | Expected content |
|---|---|
| **Meta-data** | |
| Title | Title should provide a concise description of the nature and topic of the study, including the name of the data collection method [34]. General recommendations can be found in [35]. |
| Abstract | Abstract should report objectives (state of the problem under investigation, main hypotheses); description of participants (pertinent characteristics); study method (research design, sample size, materials/methods used, outcome measured, data analysis procedures); main findings (including statistical significance levels); conclusions (beyond just results, reporting implications or applications). Adapted from [33]. |
| Keywords | Keywords should be related to controlled vocabularies. Adapted from [36]. |
| **Introduction** | |
| Problem | Introduction should describe the importance of the problem, including an overview of what is known about the problem, gaps in current knowledge and practical implications that make the study necessary. Adapted from [34]. <br> Introduction should include a succinct review of relevant scholarship, including relation to previous work. Included references should be relevant to the problem studied, and cited in accordance with content and context. Self-citation should be justified. Adapted from [37]. |
| Objective(s) | Introduction should report a statement framed as one or more research questions, purposes, goals, or objectives that should set readers' expectations for the methods, findings and discussion sections of the manuscript. Adapted from [34]. |
| **Materials and methods** | |
| Participants | A participants section should be reported, indicating: inclusion and exclusion criteria (if any); major demographic characteristics as well as important research-specific characteristics; recruitment procedure; settings, locations and dates for data collection; consent and retribution made to participants; institutional review board agreements; ethical standards met and safety monitoring (if any); intended and achieved sample size if different from intended; determination of sample size (power analysis or methods used to determine the number). Adapted from [33]. |
| Data collection | Authors should describe in detail their data collection design and method(s) and justify them in relation to the research question(s). The authors should describe all instruments, guides, and protocols, including their development and cite relevant literature, theories or conceptual frameworks as appropriate. Adapted from [34]. |

(*Continued*)

**Table 4.** (Continued)

| Topic | Expected content |
|---|---|
| **Data analysis** | Materials and methods should include a data analysis section describing the analytic process so that readers can follow the logic of inquiry from the research question(s) to the analysis and findings. The authors should cite the guiding literature and describe their processes in sufficient detail so readers can judge the extent to which the processes align with the guiding approach. If modification to or deviations from the guiding approach occurred, the authors should explain and justify these modifications. Adapted from [34].<br>Data analysis section should report: the variables measured and their nature (intensity, citation rate, etc.); the planned data diagnostics (criteria for post-data collection exclusions of participants); the criteria for deciding when to infer missing data and methods used for imputation of missing data; the definition and processing of statistical outliers; the analyses of data distributions; the data transformation to be used; the statistics methods used (including details of the models and references to the appropriate literature if required); the data analysis software. Adapted from [33].<br>Before engaging in statistical inferences, level of expected statistical probability (e.g. $p<0.05$, $p < .01$) should be established on the basis of reasonable knowledge of the phenomena under investigation and the caution necessary for interpreting comparisons [38]. |
| **Results** | |
| **Synthesis** | Results should report information detailing the statistical and data-analytic methods, including: missing data; characterization of the data (n, means, standard deviations, etc.); inferential statistics (including exact p-values, minimally sufficient set of statistics needed to construct the tests: dfs, mean square effects, mean square error, etc.); reporting of any problems with statistical assumptions and/or data distributions that could affect the validity of findings. Adapted from [33]. |
| **Findings** | Results should report evidence to substantiate the more general and abstract concepts or inferences presented as findings. Authors should report counter-examples and concrete details related to their findings. Judicious use of tables and figures can help communicate such findings Adapted from [34]. |
| **Quality of measurements** | "Just because one obtains a graphical display or a series of tables with associated statistical significance does not mean it has any meaning or external validity" [39].<br>For new methods, results should provide validity evidence (either directly in the study itself, e.g., via pilot testing, or indirectly based on previous research) supporting the use of the given measurement instruments for the intended construct interpretations [38]. More on validity can be found in [40].<br>Results should also report estimates related to the reliability of measures (reliability estimates from other studies should only be used for comparison purposes). Adapted from [32, 37, 39]. More on reliability can be found in [40]. |
| **Discussion** | |
| **Support of original hypotheses** | Discussion should begin with a short summary of the main findings as a remainder for the readers helping them assess whether the subsequent interpretation and implications formulated are supported by the findings. Adapted from [34]. It should provide a statement of support or non-support for all hypotheses. Adapted from [33]. |
| **Connection to prior works** | Discussion should elaborate on similarities and differences between reported results and work of others. Adapted from [33]. |
| **Interpretation & limitations** | Discussion should provide an interpretation of the results and elaboration on findings in relation to the study purpose. Specific elements, decisions or events of the study that could influence interpretation should be identified. Adapted from [34]. Authors should also take into account: sources of potential bias; imprecision of measurement protocols; overall number of tests; adequacy of sample size [33]. |
| **Contribution to the field** | Discussion should consider contribution to the field [34], generalizability of the findings (taking into account target population and other contextual issues) and implications for future research [33]. |
| **Transparency** | |
| **Declaration of interest** | This section should identify (if applicable) any real or potential conflicts of interest that might have influenced or could appear to have influenced the research. Authors should also explain how these conflicts were managed in the conduct of the study, and describe the potential impact on study findings and/or conclusions [34]. |

(*Continued*)

**Table 4.** (Continued)

| Topic | Expected content |
|---|---|
| Funding | This section should describe any sources of funding and other support for the study and the role of funders in data collection, data analysis and reporting, if applicable [34]. |
| Contributions | This section should report the roles played by each author of the article, using Contributor Roles Taxonomy. [41]. |

other purpose, the predetermined answers were "yes", "no" or "not applicable". The result is summarized in Table 5.

The two authors will independently fill in the data extraction form materialized by an Excel sheet (that will be included as a supplementary material in the final review). As the nature of the answers is mainly objective, the differences in reporting will be resolved by checking the article until agreement between the 2 reviewers.

**Table 5. Extraction form based on topic-related questions.** Items marked with an asterisk (*) are mandatory (except for statistical oriented articles) for the article to be considered for quality appraisal (see section 2.4.5).

| Item | Question<br>*Predetermined answers* |
|---|---|
| **Meta-data** | |
| DOI | What is the digital object identifier of the article?<br>*As reported in Scopus/WOS.* |
| Title | What was title of the article?<br>*As reported in Scopus/WOS.* |
| Authors | Who are the authors of the article?<br>*As reported in Scopus/WOS.* |
| Abstract | What is the content of the abstract of the article?<br>*As reported in Scopus/WOS.* |
| Year | What is the year of publication of the article?<br>*As reported in Scopus/WOS.* |
| Source title | What is the title of the peer-reviewed journal in which the article is published?<br>*As reported in Scopus/WOS.* |
| Keywords | What are the keywords reported by the authors?<br>*As reported in Scopus/WOS.* |
| Language | What is the language of the article?<br>*As reported in Scopus/WOS.* |
| Subject/research area | What is the subject area of the journal?<br>*As reported in Scopus/WOS.* |
| Number of references | How many references are cited by the article?<br>*As reported in Scopus/WOS.* |
| Number of citations | How many references cite the article at the moment of the review?<br>*As reported in Scopus/WOS.* |
| Open access to the manuscript | What is the type of open access ?<br>*As reported in Scopus/WOS.* |
| Open access to the data | Does the reader have access to the data, in a public repository or data paper? (extends "Open access to the manuscript")<br>*Yes/no.* |
| Open access to the source code | Does the reader have access to the source code used for data analysis? (extends "Open access to the manuscript")<br>*Yes/no.* |
| **Introduction/Problem** | |

(*Continued*)

**Table 5.** (Continued)

| Item | Question<br>*Predetermined answers* |
|---|---|
| **Review of scholarship**[*] | Did the introduction include an overview of what is known of the problem based on a review of the scholarship (related to temporal sensory analysis)?<br>*Yes/no.* |
| **Relevance** | Did the introduction identify gaps in current knowledge and/or practical implications that make the study necessary?<br>*Yes/no.* |
| **Introduction/Objectives** | |
| **Objective(s)**[*] | What were the objectives of the research?<br>*As reported by the authors/not reported.* |
| **Area of knowledge** | What was the area of knowledge produced by the research?<br>*Methodological/product oriented/etc. (deduced from objective(s))* |
| **Materials and methods/participants** | |
| **Selection criteria** | Were the selection criteria of the participants reported?<br>*Yes/no/not applicable.* |
| **Recruitment modalities** | Were the recruitment modalities reported?<br>*Yes/no/not applicable.* |
| **Determination of sample size** | What was the criterion for determining sample size?<br>*Literature/power calculation/not applicable/not reported.* |
| **Number**[*] | What was the final number of participants that actually participated in the evaluation?<br>*As reported by the authors/not applicable/not reported.* |
| **Demographics** | Were the characteristics of the participants (at least one characteristic: age, gender, frequency of consumption, etc.) reported?<br>*Yes/no/not applicable.* |
| **Country** | In which country did the experiment take place?<br>*As reported by the authors (or deduced from context of the experiment, but not by authors affiliation)/not reported/not applicable.* |
| **Location** | Where did the data collection take place?<br>*Lab/home/not applicable/etc.* |
| **Ethics review board** | Did the authors report an approval by an appropriate ethics review board?<br>*Yes/No/Not applicable* |
| **Participant consent** | Did the authors report participants consent?<br>*Yes/No/Not applicable* |
| **Nature of the compensation** | Did the authors report if there was or not a compensation for the participants?<br>*Yes/No/Not applicable* |
| **Materials and methods/products** | |
| **Description**[*] | Did the authors report relevant information about the food products? (brand, recipe, composition, etc.)<br>*Yes/no/not applicable.* |
| **Type** | What was the type of product? (deduced from product description)<br>*Commercial/model (prepared by the experimenter)/not reported/not applicable.* |
| **Food category** | What was the category of the food products? (deduced from product description) [42]<br>*Wine/chocolate/not applicable/etc.* |
| **Physical state** | What was the physical state of the food products? (deduced from product description)<br>*Solid/semi-solid/liquid/not applicable.* |
| **Serving conditions** | Did the authors report the relevant information (portion size, container, temperature, light, etc.) about the serving conditions?<br>*Yes/no/not applicable.* |
| **Information given to the participants** | If the product is not blindly evaluated, what was the information given to the participants?<br>*None/Brand/Allegation/Price/Package/etc.* |

(*Continued*)

**Table 5.** (Continued)

| Item | Question<br>*Predetermined answers* |
|---|---|
| **Number**[*] | How many different products/samples were evaluated?<br>*As reported by the authors/not reported/not applicable.* |
| **Materials and methods/attributes** | |
| **Selection** | How were the attributes selected for the study?<br>*By the panel/by another panel/literature/not reported/not applicable/etc.* |
| **Description**[*] | What were the names of the attributes?<br>*As reported by the authors/not reported/not applicable.* |
| **Sensory modalities** | What were the sensory modalities evaluated? (deduced from description)<br>*Basic taste/flavor/texture/mouthfeel/not applicable/etc.* |
| **Definitions** | Were attribute definitions presented to the participants?<br>*Yes/no/not applicable.* |
| **References** | Were attribute references presented to the participants?<br>*Yes/no/not applicable.* |
| **Number** | How many attributes were evaluated?<br>*As reported by the authors/not reported/not applicable.* |
| **Materials and methods/research design** | |
| **Object(s) of comparison** | What was the object of interest?<br>*Method/product/intake/subject/not applicable/etc.* |
| **Temporal unit** | What was the temporal unit of the measures?<br>*Within-intake/between intakes/etc.* |
| **Study design**[*] | What was the study design used for comparing the experimental units?<br>*Within balanced/within unbalanced/between/factorial/not applicable/not reported/etc.* |
| **Product order** | What was the experimental design defining the rank of presentation of the samples?<br>*Balanced/randomized/not reported/not applicable/etc.* |
| **Attribute order** | What was the experimental design defining the rank of presentation of the attributes?<br>*Balanced/randomized/not reported/not applicable/etc.* |
| **Material and methods/data collection** | |
| **Temporal method(s)**[*] | What was the name of the temporal sensory method/variant involved?<br>*TDS/TDS intensity/TI/TCATA/TCATA fading/etc.* |
| **Other measures** | Was there other information collected with the temporal sensory data?<br>*Liking/physicochemical measures/none/etc.* |
| **Training** | How were the participants introduced to the method?<br>*Familiarization/training (hours)/not reported/not applicable.* |
| **Type of panel**[*] | What was the type of panel? (depending on the training of the participants, [43])<br>*Trained/semi-trained/expert/consumer/not reported/not applicable.* |
| **Instructions** | Did the authors report the instructions given to the participants?<br>*Yes/no/not applicable.* |
| **Warm-up** | Did the tasting include a warm-up product prior to the evaluation of the samples?<br>*Yes/no/not applicable.* |
| **Software** | Which was the software used for temporal sensory data collection?<br>*As reported by the authors/not reported/not applicable.* |
| **Number of evaluations** | How many times were the samples evaluated (replicates)?<br>*As reported by the authors/not reported/not applicable.* |
| **Standardization of the tasting** | How was the tasting standardized?<br>*As reported by the authors/not reported/not applicable.* |
| **Duration of the tasting** | How long did the standardized tasting last?<br>*Fixed duration/free duration (time)/not reported/not applicable.* |
| **Material and methods/data analysis** | |

(*Continued*)

**Table 5.** (Continued)

| Item | Question<br>*Predetermined answers* |
|---|---|
| **Justification of data selection** | Did the authors justify data selection (subjects, products, attributes), if any?<br>*Yes/no/not applicable.* |
| **Data transformation** | How was data transformation (if any) carried out?<br>*Time standardization/periods/none/etc.* |
| **Variables*** | What were the analyzed variables?<br>*Durations of dominances/citation rates/intensities/not reported/etc.* |
| **Statistics*** | What were the statistical analyses used?<br>*Curves/PCA/ANOVA/not reported/etc.* |
| **Alpha** | Was the level of expected statistical probability reported (previously to the results)?<br>*Yes/no/not applicable.* |
| **Software** | What was the software used for data analysis?<br>*R/SAS/etc.* |
| **Results/Synthesis** | |
| **Characterization of data** | Did the authors provide a characterization of the data? (missing data, descriptive statistics including n, mean, standard deviations, etc.)<br>*Yes/no/not applicable.* |
| **Inferential statistics** | Did the authors report inferential statistics parameters when comparing objects of interest (products, methods, etc.)? (p-values, dfs, mean square effects, mean square error, etc.)<br>*Yes/no/not applicable.* |
| **Results/Findings** | |
| **Main findings*** | What were the main findings related to temporal sensory analysis?<br>*As reported by the authors/not reported.* |
| **Results/Quality of measurements** | |
| **Validity** | Did the authors report validity evidence supporting the use of the method?<br>*Yes/no/not applicable.* |
| **Reliability** | Was a measure of reliability included?<br>*Yes/no/not applicable.* |
| **Discussion** | |
| **Answer to research question** | Did the authors provide an answer to their research question?<br>*Yes/no.* |
| **Connection to prior works** | Did the authors report connections to prior works?<br>*Yes/no.* |
| **Interpretation*** | Did the authors provide an interpretation of the results in relation to the study purpose?<br>*Yes/no.* |
| **Limitations** | Did the authors report elements that could have biased the result or influenced the interpretation?<br>*Yes/no.* |
| **Contribution to the field** | Did the authors discuss the contribution to the field (generalizability of the findings and/or implications for future researches)?<br>*Yes/no.* |
| **Transparency** | |
| **Declaration of interest** | Were the potential sources of influence on study conduct and conclusions reported?<br>*Yes/no.* |
| **Funding** | Were the sources of funding and other support reported?<br>*Yes/no.* |
| **Contributions** | Were the contributions of the authors included?<br>*Yes/no.* |

**2.4.4. Evaluation of completeness of data form.**  Once the data extraction for an article is filled, its completeness will be evaluated. Answers "no" and "not reported" will count as 0 while other answers will count as 1 for the purpose of summing across topics of Table 4. Nine completeness indicators will be computed in this way: "Introduction" (0 to 4), "Materials and methods/participants" (0 to 10), "Materials and methods/products" (0 to 7), "Materials and methods/attributes" (0 to 6), "Materials and methods/research design" (0 to 5), "Material and methods/data collection" (0 to 10), "Material and methods/data analysis" (0 to 6), "Results" (0 to 5), "Discussion" (0 to 5).

The evaluation of the completeness will not require any subjective judgment as it will be based on consensual answers of Table 5. Thus, the completeness indicators will be automatically computed based on the data extraction form thanks to an Excel formula.

**2.4.5. Critical appraisal.**  Only peer-reviewed articles will be included in the final scoping review. Nonetheless, some authors stress the importance of a thorough quality assessment in scoping reviews [44, 45]. To achieve this objective, 7 quality indicators (QI) were chosen according to [46]. It should be noted that only the articles that have the information identified as mandatory (see Table 5) will be evaluated in this stage. The articles that do not comply with this condition will be retained but marked as "not evaluated".

To help the reviewer in their assessments, the QI were divided in topic-related questions summarized in Table 6.

The reviewers' involvement will be added for the sake of transparency about the neutrality of the appraisal (it will be answered "yes" if at least one the reviewers is or was personally involved in the work or with the authors, considering this as potential bias in the quality appraisal).

The two reviewers (R1 and R2) will independently answer to all topic-related questions, each having 3 possible answers, "yes", "no" and "can't tell" (Critical Appraisal Skills Program [47]). Then, the final judgement will be obtained as follows:

**Step 1. For each topic-related question:**

```
R1 "yes" + R2 "yes" = "yes"
R1 "yes" + R2 "can't tell" = "probably yes"
R1 "can't tell" + R2 "can't tell" = "can't tell"
R1 "no" + R2 "can't tell" = "probably no"
R1 "no" + R2 "no" = "no"
R1 "yes" + R2 "no" = "disagreement"
```

Reasons for giving "no" as an answer will have to be justified.

**Step 2. For the Quality Item the least favorable evaluation among the topic-related final evaluations will be retained.**

For example, if reviewer 1 answered "yes" to QI6a while reviewer 2 answered "can't tell", QI6a final evaluation will be "probably yes". If the 2 reviewers answered "can't tell" to QI6b, QI6a final evaluation will be "can't tell". Considering the answers to parts a and b of QI6, its final evaluation will be "can't tell".

No overall score quality evaluation (taking into account the seven QI) will be made, and all articles will be included whatever their score. Indeed, these scores will only reflect a quality level related to the research question of this review, and the final appraisal will be to the discretion of the future readers of the scoping review. The evaluation of each article consisted in as a 3-steps procedure (read below):

1. Extraction of information (63 items) using the extraction form

2. Automatic evaluation of the completeness of the reporting (9 indicators)

**Table 6. Indicators of quality derived from [46].**

| Quality indicator (QI) | Topic-related questions and related items |
|---|---|
| QI1 –Clear research question? | QI1a –Was the introduction detailed enough to give an overview of the problem comprehensive for a competent but non-expert reviewer? ("Review of scholarship"). <br> QI1b –Was the necessity of the study justified from the perspective of a competent but non-expert reviewer? ("Relevance"). <br> QI1c –Did the stated objectives set readers' expectations for the methods, findings and discussion? ("Objective(s)", "Area of knowledge"). |
| QI2 –Appropriate participants? | Were the participants appropriate for answering to the research question? ("Selection criteria", "Determination of sample size", "Number", "Location", "Demographics"). |
| QI3 –Appropriate design & data collection? | QI3a –Were the research design parameters appropriate to answer the research question? ("Object(s) of comparison", "Temporal unit", "Study design", "Product order", "Attribute order") <br> QI3b –Was the choice of all the methods justified by literature and/or appropriate to answer the research question? ("Temporal method(s)", "Other measures") <br> QI3c –Was the implementation of the method appropriate to answer the research question? ("Training", "Type of panel", "Instructions", "Number of evaluations", "Standardization of the tasting", "Duration of the tasting") <br> QI3d –Was the data collection process described in a way that makes the experiment reproducible? |
| QI4 –Appropriate data analysis? | QI4a - Was the analysis process made according to the literature or justified in case of a new approach? ("Data selection", "Data transformation", "Variables", "Statistics", "Alpha", "Software") <br> QI4b - Was the data analysis process described in a way that makes it reproducible? |
| QI5 –Claims supported by evidence? | QI5a - Did authors make an appropriate synthesis of the results, including a judicious use of tables and figures allowing to characterize raw data and statistics? ("Characterization of data", "Inferential statistics") <br> QI5b - Did authors report sufficient data and substantial evidence to support the findings? ("Main findings", "Validity", "Reliability") |
| QI6 –Integrated interpretations and conclusions? | QI6a - Did authors appropriately discuss their findings in relation to the study objectives and prior works? ("Support of original hypotheses", "Connection to prior works", "Interpretation") <br> QI6b - Did authors objectively report sufficient elements pro and against their arguments? ("Limitations") |
| QI7 –Useful contribution? | Did authors discuss the generalizability of their findings and/or implications for future research? ("Contribution to the field") |

3. Critical appraisal (7 indicators)

Fig 2 summarizes the evaluation process that was refined and tested on the 50 test articles. The modifications to this evaluation process, if any, will be described in the final scoping review.

## 3. Results

Tables and figures will be produced to summarize most of the extracted data presented in Table 5. A narrative synthesis organized into themes reflecting the scoping review objectives will also be presented. Both descriptive statistics and narrative synthesis will be supported by a qualitative analysis.

## 4. Discussion

The proposed scoping review resulting from the present protocol aims at summarizing how multi-attribute temporal descriptive methods have been implemented, used and compared in

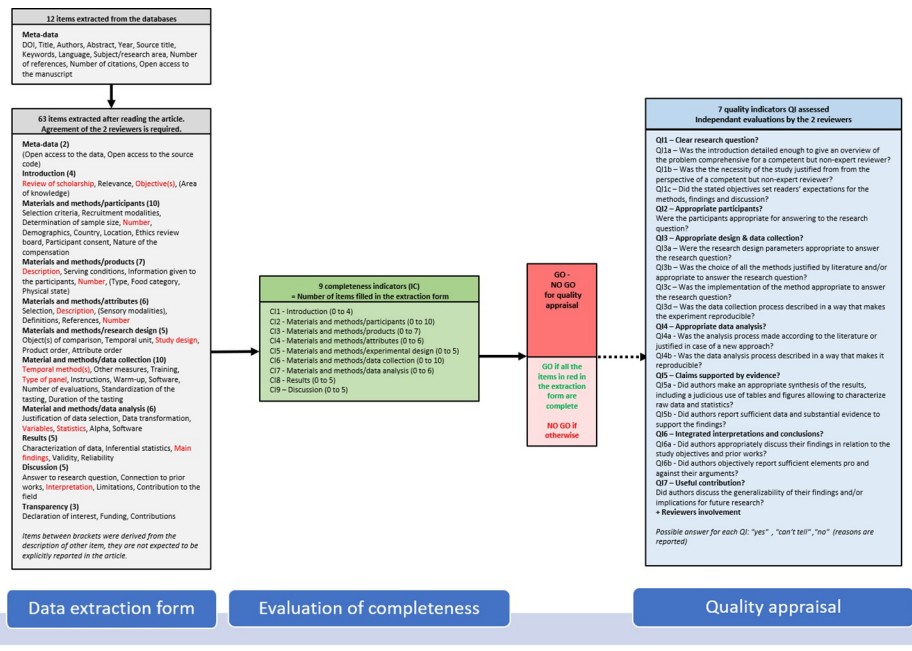

**Fig 2. Summary of the process of evaluation.**

sensory analysis. This review will help the sensory analysts choose the appropriate method according to their needs (e.g. product description, product discrimination, oral processing behavior) and adopt the best practices. Moreover, this review will allow to identify the areas where additional research and/or validation is necessary.

This protocol presents a detailed methodology for conducting the review. The publication of this protocol presents several advantages. The protocol has been reviewed thanks to expert feedbacks, ensuring its transparency and validity. The data extraction form and the quality appraisal flowchart can be adapted and reused for other areas of research, particularly in a young science such as sensory and consumer science. This protocol could also be used as checklists to ensure no important information has been forgotten when writing new articles in this field. Indeed, the training exercise on the first 50 articles showed a great heterogeneity in the way the details of the studies are reported. This protocol would help standardizing the way the results of research are reported, which is important with the emergence of open science. Moreover, the authors would also point out that despite the fact that only peer-reviewed articles have been included, several studies do not meet the quality requirements that should be expected. This could be due to a lack of information or validated sources in the field. This protocol could also be used as a guideline for reviewing research manuscripts, or at least to point out the need for defining consensual criteria among journals. Finally, this protocol will promote the use of systematic reviews in science to inform the debate and improve the quality of future research.

Limitations in this protocol have to be reported. The research strategy largely depends on the name of the temporal methods. Therefore, the authors may have missed new or little-known temporal methods. Moreover, in order to limit the number of retrieved articles by the database queries, only specific subject areas (Scopus) or research areas (WOS) have been investigated. Again, it could result in missing articles published in annex research fields such as mathematics or computer science. However, these two limits are not that serious, because it is

likely that the missed articles cite or have been cited by at least once one of the articles retrieved through the database queries. The additional references identified this way will be reported in the PRISMA diagram. The research has been limited to articles published in English language. A first look on the geographic distribution of the included studies suggests that it is not a concern. Only peer-reviewed articles have been considered for inclusions, which is debatable. This choice was made for the sake of feasibility and reproducibility. Moreover, it is unlikely that the excluded articles could have met the quality criteria, and the corpus of included publications is supposed large enough to answer to the research questions. Finally, only two databases have been queried. In a first time, Google Scholar was considered, but except for gray literature this database did not bring additional references. This seems to suggest that it was sufficient, considering all the limitations mentioned above.

## Author Contributions

**Conceptualization:** Michel Visalli, Mara Virginia Galmarini.

**Investigation:** Mara Virginia Galmarini.

**Methodology:** Michel Visalli, Mara Virginia Galmarini.

**Visualization:** Michel Visalli.

**Writing – original draft:** Michel Visalli, Mara Virginia Galmarini.

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
