## [Decision Letter · Decision Letter 0]

6 Jun 2022

PONE-D-22-05542

Multi-attribute temporal descriptive methods in sensory analysis applied in food science: protocol for a scoping review

PLOS ONE

Dear Dr. Visalli,

Thank you for submitting your manuscript to PLOS ONE. After careful consideration, we feel that it has merit but does not fully meet PLOS ONE’s publication criteria as it currently stands. Therefore, we invite you to submit a revised version of the manuscript that addresses the points raised during the review process.

I am so sorry for late processing. Many researchers I asked to be as reviewers were your collaborators, so I cannot find so much candidates for reviewing. I could find some reviewers, but they could not accept or reply their review comments. Finally, I can get comments from a reviewer! I apologize  for the delay.

I also reviewed your manuscript and cannot find any reason for not accepting your manuscript as an article of PlosOne. I am happy for accepting your manuscript after minor revision. Please find comments from reviewer 1. I am looking forward to reading your revision soon.

We look forward to receiving your revised manuscript.

Kind regards,

Nobuyuki Sakai, Ph.D.

Academic Editor

PLOS ONE

**Journal requirements:**

https://journals.plos.org/plosone/s/file?id=ba62/PLOSOne_formatting_sample_title_authors_affiliations.pdf"

Reviewers' comments:

Reviewer's Responses to Questions

**Comments to the Author**

1. Does the manuscript provide a valid rationale for the proposed study, with clearly identified and justified research questions?

Reviewer #1: Yes

2. Is the protocol technically sound and planned in a manner that will lead to a meaningful outcome and allow testing the stated hypotheses?

Reviewer #1: Yes

3. Is the methodology feasible and described in sufficient detail to allow the work to be replicable?

Reviewer #1: Yes

4. Have the authors described where all data underlying the findings will be made available when the study is complete?

Reviewer #1: Yes

5. Is the manuscript presented in an intelligible fashion and written in standard English?

Reviewer #1: Yes

6. Review Comments to the Author

You may also provide optional suggestions and comments to authors that they might find helpful in planning their study.

Reviewer #1: The manuscript presents a very interesting integrative review with an original and relevant protocol to conduct a scoping review on multi-attribute temporal descriptive methods in sensory analysis. It is well written and the plan is clear. The tables are particularly very relevant and complete. Congratulation! I find very relevant to publish it. My main concern is about the end of the paper. It could be good to better highlight how this contribution could be used for future publications in temporal descriptive studies and to better show the interest and main outcomes.

Remarks / questions:

Line 43 : “oral processing” could be more integrative than “intake”?

Line 46 : what about the quantification of changes in perception? This review is focused only on description and not quantification?

Table 1 : how was chosen the order of presented methods in the table? Maybe it could be clearer to use the same order as in the text?

Concerning attribute generation, for some methods (progressive profiling, TDS, sequential profiling), it could be better to use “after panel consensus” than pre-determined by panel leader?

Line 63: maybe add the problem of signature effects as other limits?

Line 73: add references

Lines 80-90: I'm embarrassed by "newer methods": maybe be more precise?

Line 100: replace he by it

Line 102: it could be interesting to mention when their application is also recommended?

Line 105: in this part, it could good: “which type or category of products were studied?

Table 5: very very interesting!

Part 4 discussion: I suggest to complete with some perspectives and further uses of these results.

7. PLOS authors have the option to publish the peer review history of their article (what does this mean?). If published, this will include your full peer review and any attached files.

Reviewer #1: No

---

## [Author Response · Author response to Decision Letter 0]

8 Jun 2022

Comment to editor and reviewer

After testing the protocol on several articles, the authors found some definitions to be unclear. Therefore, minor changes have been done in Table 5. These are colored in blue.

Reviewer #1: The manuscript presents a very interesting integrative review with an original and relevant protocol to conduct a scoping review on multi-attribute temporal descriptive methods in sensory analysis. It is well written and the plan is clear. The tables are particularly very relevant and complete. Congratulation! I find very relevant to publish it. My main concern is about the end of the paper. It could be good to better highlight how this contribution could be used for future publications in temporal descriptive studies and to better show the interest and main outcomes.

Thank you for your helpful comments. Please find below our answers to their questions. Corrections in the manuscript have been colored in blue.

Remarks / questions:

Line 43 : “oral processing” could be more integrative than “intake”?

The suggested change has been made in the revised version (line 43). 

Line 46 : what about the quantification of changes in perception? This review is focused only on description and not quantification?

The suggested change has been made in the revised version: "Thus, several temporal methods have been developed in the past 50 years trying to capture, study, describe and quantify these changes in perception" (line 46).

Table 1 : how was chosen the order of presented methods in the table? Maybe it could be clearer to use the same order as in the text?

Methods were presented in a chronological order. This was precised of the revised version (line 57).

Concerning attribute generation, for some methods (progressive profiling, TDS, sequential profiling), it could be better to use “after panel consensus” than pre-determined by panel leader?

The reviewer is correct, this was not clear in the original version. Categories of attribute generation have been changed to: "Determined before the test (by the panel leader, by consensus, etc.)" and " Determined during the test (Free-Comment)".

Line 63: maybe add the problem of signature effects as other limits?

The problem of the signature effect has been added in the revised version: "Moreover, the "signature" effect [14] (evaluators have a characteristic shape of the curve) requires a higher training to reduce variability and obtain curves that respond to product characteristics and not to individual differences, resulting also in panellist fatigue [13]." (lines 64-66).

Line 73: add references

References have been added after each method both in table and text.

Lines 80-90: I'm embarrassed by "newer methods": maybe be more precise?

In the revised version, “Newer methods” was replaced by “methods developed after this” (line 84).

Line 100: replace he by it

The suggested change has been made in the revised version (line 104).

Line 102: it could be interesting to mention when their application is also recommended?

The sentence has been rephrased to “It has not been clearly established yet in which situations methodologies provide equivalent information and when their application is or is not recommended.” (line 106).

Line 105: in this part, it could good: “which type or category of products were studied?

This sentence (now italicized) is the original quote from a researcher referring to the need to validate the current sensory methods. Using the proposed protocol might help finding out which products are better described by the different methods. In the section "method and protocol" we have added the question "which product categories were evaluated using the methods?" to make sure that we have this information (line 158).

Table 5: very very interesting!

We thank the reviewer for this comment.

Part 4 discussion: I suggest to complete with some perspectives and further uses of these results.

The discussion has been partially rewritten to make the perspectives more explicit (lines 422-449).

---

## [Editor Report · Decision Letter 1]

22 Jun 2022

Multi-attribute temporal descriptive methods in sensory analysis applied in food science: protocol for a scoping review

PONE-D-22-05542R1

Dear Dr. Visalli,

We’re pleased to inform you that your manuscript has been judged scientifically suitable for publication and will be formally accepted for publication once it meets all outstanding technical requirements.

Kind regards,

Nobuyuki Sakai, Ph.D.

Academic Editor

PLOS ONE
---

## [Editor Report · Acceptance letter]

27 Jun 2022

PONE-D-22-05542R1 

Multi-attribute temporal descriptive methods in sensory analysis applied in food science: protocol for a scoping review. 

Dear Dr. Visalli:

I'm pleased to inform you that your manuscript has been deemed suitable for publication in PLOS ONE. Congratulations! Your manuscript is now with our production department. 

Kind regards, 

on behalf of

Dr. Nobuyuki Sakai 

Academic Editor

PLOS ONE